# The Complexities of Metastasis

**DOI:** 10.3390/cancers11101575

**Published:** 2019-10-16

**Authors:** Beatriz P. San Juan, Maria J. Garcia-Leon, Laura Rangel, Jacky G. Goetz, Christine L. Chaffer

**Affiliations:** 1The Kinghorn Cancer Centre, Garvan Institute of Medical Research, Darlinghurst 2010, Australia; b.perez@garvan.org.au (B.P.S.J.); l.rangel@garvan.org.au (L.R.); 2St Vincent’s Clinical School, University of New South Wales Medicine, University of New South Wales, Darlinghurst 2010, Australia; 3INSERM UMR_S1109, Tumor Biomechanics, 67000 Strasbourg, France; m.garcia@inserm.fr; 4Université de Strasbourg, 67000 Strasbourg, France; 5Fédération de Médecine Translationnelle de Strasbourg (FMTS), 67000 Strasbourg, France

**Keywords:** metastasis, heterogeneity, plasticity, epithelial-to-mesenchymal transition, biomechanics, circulating tumor cells (CTCs), extracellular vesicles, metastatic niche, epigenetics, CTC-clusters

## Abstract

Therapies that prevent metastatic dissemination and tumor growth in secondary organs are severely lacking. A better understanding of the mechanisms that drive metastasis will lead to improved therapies that increase patient survival. Within a tumor, cancer cells are equipped with different phenotypic and functional capacities that can impact their ability to complete the metastatic cascade. That phenotypic heterogeneity can be derived from a combination of factors, in which the genetic make-up, interaction with the environment, and ability of cells to adapt to evolving microenvironments and mechanical forces play a major role. In this review, we discuss the specific properties of those cancer cell subgroups and the mechanisms that confer or restrict their capacity to metastasize.

## 1. Cancer Cell Heterogeneity: A Hierarchical Matter?

### 1.1. Cancer Origin and Evolution

A normal cell transforms into a cancer cell by accrual of multiple genetic mutations over time, which ultimately lead to uncontrolled cellular proliferation. Genetic drift may arise from a combination of germline or spontaneous mutations, exposure to environmental carcinogens, genome rearrangements, and/or increased genome instability [1]. Those genetic changes can subsequently impact a cancer cell’s epigenetic landscape by changing chromatin regulatory machinery or by aberrant expression of transcription factors that normally drive cellular differentiation and specify cellular fate [2]. To add to that complexity, the genomic/epigenomic drivers of a cancer can change over time. Standard-of-care treatment for most solid tumors comprises a series of aggressive chemotherapies that, in combination with aberrant cancer cell divisions and fluctuating microenvironmental landscapes, create opportunity for cancer cells to further mutate, adapt, and evolve, often toward a more aggressive phenotype. In this way, genetic and epigenetic modifications create phenotypic and functional heterogeneity [3] that fuel tumor progression and, consequently, represent a major therapeutic obstacle [4].

### 1.2. A Cancer Cell Hierarchy

Notwithstanding the genetic component to cancer development and progression, it is also well established that epigenetic mechanisms can create functional heterogeneity in genetically identical cancer cells, which is fundamentally important to tumor growth and metastasis. That notion is solidified in the idea that genetically identical cancer cells can be hierarchically organized according to phenotype, in this case, tumor-initiating potential [5,6,7,8,9]. At the top of the hierarchy sit the aggressive cancer stem cells (CSCs, or tumor-initiating cells), which, in a manner akin to stem cell divisions in normal tissues, self-renew to maintain the tumor-initiating cell pool or divide asymmetrically to produce non-tumor-initiating cell progeny (Figure 1). The balance between self-renewal and differentiation is determined by a combination of cell-intrinsic and environmental factors, which can dynamically impact cellular heterogeneity observed within a tumor. Generally, a higher percentage of tumor-initiating cells is associated with more aggressive and metastatic tumors [10,11]. With the unique capacity to fuel tumor growth, to metastasize, and to resist therapeutic treatment, attempts to better identify and functionally characterize those aggressive cells are of great interest.

The origins of tumor-initiating cells are not yet clearly defined; however, it has been hypothesized that they may arise via oncogenic transformation of normal tissue stem or progenitor cells [12,13,14]. Alternatively, tumor-initiating cells may also arise via reversion of non-tumor-initiating cancer cells into a tumor-initiating cell state [7,10,15]. That idea is conceptually important as it implies that tumor-initiating cells can be continually replenished throughout tumorigenesis. Moreover, it provides a mechanism for non-aggressive tumors to transition toward more aggressive and metastatic disease. Accordingly, the characterization of signaling mechanisms that generate and maintain highly tumorigenic, metastatic, and chemotherapy-resistant tumor-initiating cells should provide novel avenues for therapeutic design. In that regard, the development of new technologies, such as single-cell sequencing [16,17] and barcoding-based functional assays [9,18,19,20], applied to clinically relevant models, should be able to address these issues in the near future. 

### 1.3. CD44: Defining Aggressive Cancer Cells

The use of membrane-anchored protein markers to distinguish subpopulations of aggressive cancer cells has proven a useful tool in cancer research [21,22,23,24,25]. In a wide variety of solid tumors, including those of breast [6], gastric [26], pancreas [27], ovary and colon [28,29,30], and also in blood malignancies [31], residence in, or transition into the aggressive tumor-initiating cell state can be monitored by high expression of the cancer stem cell marker CD44—where the nomenclature CD44^Hi^ represents cells enriched for that aggressive cancer cell phenotype [32,33,34,35,36]. The CD44 molecule can exist in a variety of splice isoforms that are functionally important. Recent findings link the expression of different CD44 isoform variants with cancer progression and specific tumor cell features, including pro-survival signaling [37], cellular reprogramming [38], acquisition of migratory capacity [39], and tumor initiation [40,41,42]. CD44 can also facilitate the arrest of circulating tumor cells prior to extravasation [43]. Together, these findings emphasize the cellular and molecular heterogeneity that exists within cancer cell populations, which belie the power of bulk population analyses to define putative therapeutic options. 

#### 1.3.1. CD44^Lo^ versus CD44^Hi^ Cells: Epithelial versus Mesenchymal Cell States

Compared to the bulk tumor mass, the aggressive CD44^Hi^ cancer stem cell subpopulation is often associated with loss of epithelial characteristics and gain of mesenchymal traits [10,15]. Consistent with those findings, activation of the epithelial-to-mesenchymal transition (EMT) program is one means by which poorly aggressive CD44^Lo^ epithelial cancer cells gain entrance into a more aggressive CD44^Hi^ cancer stem-like state [44,45,46]. The EMT transcription factors SNAI1 (snail family transcriptional repressor 1), SNAI2, ZEB1 (zinc finger E-box binding homeobox 1), among others, are key mediators of that process [47]. Indeed, ZEB1 also drives splicing of CD44 in a manner that promotes tumorigenicity, recurrence, and drug-resistance [48,49]. Along with the acquisition of cancer stem-like traits, the EMT also increases a cancer cell’s ability to invade and migrate, promotes cancer cell spread away from the primary tumor, entrance into the circulation, and extravasation at a secondary site [50]. In line with those findings, single-cell expression analysis of disseminated tumor cells isolated from breast cancer patient-derived xenograft (PDX) models at early stages of metastatic disease display gene expression profiles consistent with the EMT [17]. Additional studies in preclinical models also establish a correlation between existence of mesenchymal CSC populations and metastatic burden, and that inhibiting EMT-transcription factor expression abolishes tumor-initiation and metastatic potential of aggressive cancer cells [51,52]. Moreover, loss of an epithelial phenotype and gain of mesenchymal features correlates with poor clinical outcome in some tumor types [53,54,55,56,57,58].

#### 1.3.2. Novel Markers to Define Metastatic Cells

The search for additional markers to refine the aggressive cancer stem cell population has revealed that the CD44^Hi^ cancer cell compartment is heterogeneous and encompasses a variety of phenotypic cell states [6,59,60]. For example, expression of the marker CD24 has been used to distinguish between different cancer cell phenotypes, where enhanced tumor-initiating potential correlates with residence in a CD44^Hi^CD24^Lo^ state and the CD44^Hi^CD24^Hi^ cell state is further associated with tolerance to chemotherapy [61]. In addition, a recent study showed a novel role for integrin β4 (CD104) in the regulation of cell transitions across the epithelial–mesenchymal spectrum, where CD44^Hi^CD104^+^ cells reside in a more epithelial state than their CD44^Hi^CD104^−^ counterparts [60]. That study characterized a CD104 expression ‘sweet spot’ for tumor-initiating potential that defined a CD44^Hi^CD104^+^ intermediate epithelial–mesenchymal state [60]. Furthermore, a follow-up study demonstrated that non-canonical WNT signaling drives CD44^Hi^ cells through the CD104^+^ to CD104^−^ transition with a concomitant shift from a partial-EMT state to a mesenchymal state. That phenotypic change is indeed associated with a significant decrease in tumor-initiating potential, suggesting that retention of certain epithelial characteristics, i.e., a partial-EMT state, provides optimal tumorigenicity [54,62,63,64,65,66]. 

## 2. Cancer Cell Plasticity: Shaping Metastatic Fitness

We and others have shown that CD44^Lo^ cell populations are not locked in their epithelial state, rather they can transition into the aggressive CD44^Hi^ state via activation of components of the EMT program [15,33,35,44]. Those findings suggest that poorly tumorigenic CD44^Lo^ cells may also have the intrinsic potential to seed metastases by transitioning into a CD44^Hi^ state, albeit with far more biological effort than pre-existing CD44^Hi^ cells. If true, CD44^Lo^ cells may also be present at very early stages of metastatic dissemination. Accordingly, while pre-existing CD44^Hi^ cells are highly enriched for metastatic potential, defining a tumor’s CD44^Hi^ content at one specific time point may not adequately capture the tumor’s true metastatic potential. Additionally, and although yet to be clarified, it has been suggested that certain tumor cells are more suited to sense, compute, and respond to signals from their microenvironment that initiate the EMT program [44]. Indeed, we have previously identified that tumor cells maintaining the ZEB1 promoter in a bivalent chromatin configuration are highly conducive to activating the EMT program, or part thereof. In contrast, tumor cells that maintain the ZEB1 promoter in a repressed state are less likely to undergo the EMT [44]. Together, these studies suggest that strategies designed to prevent cellular plasticity combined with strategies to eradicate existing CD44^Hi^ cells will be required to treat cancer effectively. 

## 3. The Seed, the Journey, and the Soil: The Metastatic Cascade

Metastasis is initiated when cells migrate away from the primary tumor and invade into neighboring tissue toward blood or lymphatic vessels. After vessel wall barrier transmigration (intravasation), the invasive cells, now referred to as circulating tumor cells (CTCs), are exposed to a variety of arduous conditions, including a novel microenvironment, exposure to new cell types and signals, anchorage-independent growth, and shear forces from the blood flow. As such, survival in the circulation poses an extremely harsh selection process that very few CTCs can withstand. While CTCs are indeed detected in the majority of patients with carcinoma [67,68], it has been suggested that as few as 1–4% of CTCs successfully complete the metastatic cascade and successfully form metastatic foci [67,68,69,70]. That inefficiency suggests that CTC intrinsic features likely co-operate with surrounding tumor stroma and vascular environments to determine overall metastatic success [71,72]. CTCs thus represent a minority subpopulation of a patient’s tumor, where the role of hemodynamic forces, endothelial fitness, and blood cells are capital for tuning CTC metastatic potential. CD44^Hi^ tumor-initiating cells and the EMT program endow cancer cells with the very ability to survive these arduous conditions. Indeed, studies analyzing CTCs in human patients are enriched for an EMT phenotype [73,74].

### 3.1. Entering the Circulation, Off They Go

Tumor cells invade into their surrounding tissues toward the lymphatic and/or vascular circulation as single mesenchymal or amoeboid cell types, or collectively as epithelial sheets or clumps [75,76]. A common way for tumor cells to gain access to the circulation is via disruption of tumor vasculature integrity that enables transendothelial migration. That process is enhanced in the setting of tumor-induced chronic inflammation [77], where endothelial cell integrity and selective permeability are lost [78]. Endothelial disruption is predominantly caused by tumor infiltrating leukocytes, such as neutrophils [79,80] and macrophages [81], that communicate with tumor cells to promote intravasation by facilitating angiogenesis together with the breakdown and remodeling of the extracellular matrix [82]. In fact, macrophage depletion in mice completely abrogates breast cancer metastasis. Endothelial integrity disruption also exposes extracellular matrix proteins such as von Willebrand factor (vWF), collagen, or fibronectin, which in turn, recruit and activate platelets that act in concert to further tune tumor cell intravasation [83,84] (Figure 2). Interestingly, and together with cytokines and growth factors secreted by the tumor stroma, activated platelets at tumor vessel disruption sites can directly contribute to the initial invasive phenotype of tumor cells by the release of transforming growth factor beta TGFβ [85,86]. Indeed, platelet-derived TGFβ can induce the EMT in tumor cells entering the circulation [85,87].

Besides platelets, CTCs may also tune intravasation themselves and take advantage of the endothelial microenvironment. For example, human breast cancer cells induce mesenchymal characteristics in endothelial cells, as evidenced by upregulation of smooth muscle actin (ACTA2) and fibroblast specific protein 1 (FSP1), a phenotype also detectable in human neoplastic breast biopsies. Subsequently, the altered endothelial cells display enhanced survival, migratory, and angiogenic properties and are in turn capable of improving tumor cell survival and invasiveness via the TGFβ and Notch–Jagged1 signaling pathways [88]. Indeed, Notch ligands are frequently present on tumor-associated endothelial cells [89,90,91,92], and, independently of their roles in angiogenesis [93], they can also activate Notch signaling in tumor cells, thus enhancing aggressiveness, survival, and metastasis in diverse cancers [94,95,96]. Those advantages were precisely observed in CD44^Hi^CD24^Lo/−^ CTCs [97]. Similarly, a CD133^+^ cancer-stem cell phenotype is induced by Notch signaling in colon cancer [98]. Together, these observations indicate that the stem-like CTC phenotype may be enhanced by endothelial cell crosstalk. 

### 3.2. In Transit: Better Together

#### 3.2.1. CTC Clustering

The phenotypic, morphological, and functional properties of heterogeneous tumor cell populations at the primary tumor site, may lead to differential mechanisms of tumor cell shedding into circulation. In this sense, single CTCs and/or collectively migrating clusters—ranging from two to 50 cells—are both detected within the circulation of patients with metastatic solid cancers [99,100,101,102]. Some CTC clusters have been characterized as polyclonal tumor cell groupings suggesting that 1) they may arise from different tumor masses or metastatic foci [103,104] or 2) clustering does not necessarily occur prior to departure from the primary site, but during intravasation [105,106], transit in the circulation [103,104], or at the secondary arrest site [107] (Figure 3). Recent data derived from pre-clinical murine models demonstrate that CTC clusters show a 23–50-fold increased metastatic potential over single CTCs and are known to increase in number during disease recurrence and the development of chemotherapy resistance [74,103].

The mechanisms behind a CTC cluster’s enhanced metastatic fitness are currently under investigation. One hypothesis suggests that differential expression of cell junction proteins may play a relevant role, as cell–cell junctions are important regulators of cell phenotype and function. Indeed, preserving cell–cell contacts protects clusters from anoikis [74] and enhances their survival and colony-forming potential [103,106]. For example, knockdown of the cell junction protein plakoglobin in mouse models abrogates CTC cluster formation and drastically inhibits lung metastasis [103]. Additionally, recent findings demonstrate that CTC clusters are enriched for cells with cancer stem cell-like features [74,105], whereby intercellular homotypic interactions between the cancer stem cell marker CD44 molecules enhance cluster formation [104]. Hence, intercellular cell–cell contacts within the cluster, in addition to paracrine signals, may be key to the maintenance of that aggressive stem-like cancer cell state. Furthermore, during development, loss of cell–cell junctions is an initiating step in the EMT, while maintenance of cell–cell junctions is required to preserve the embryonic stem cell state and to reprogram somatic cells into induced-pluripotent stem cells [108,109,110]. Consistent with those findings, it has recently been shown that classic binding sites for pluripotency and proliferation-associated transcription factors such as POU class 5 homeobox 1 (POU5F1/OCT4), SRY-box transcription factor 2 (SOX2), and Nanog homeobox (NANOG, are specifically hypomethylated in clustered CTCs [111] and that pharmacological dissociation of CTC clusters reverts their methylation profile and suppresses metastasis. Those findings suggest that the distinct differentiation states between single CTCs and CTC clusters, driven in part by pluripotency factors, may account for differences in their metastatic potential. The hypothesis that hypomethylation of pluripotency sites may account for the differential metastatic potential of CTC clusters versus single CTCs is supported by data demonstrating that the DNA methylation profile of CTC clusters is detected in primary breast tumors with poor prognosis [111]. However, the specific role of EMT in CTC cluster formation and the resultant enhanced metastatic fitness remains unclear. For example, it has been shown that CTC clusters encapsulated by tumor-induced blood vessels are highly metastatic by a Slug/Snail-independent mechanism [112]. Furthermore, another report by using quantitative 3D histology at the cancer–host interface revealed that collective migration is the predominant mechanism of cancer cell invasion, positioning single cell migration as an extremely rare event [113]. These findings suggest that CTC-extrinsic mechanisms, such as vascular patterning during tumor progression, can influence CTC clustering and shedding without a compulsory phenotypic change toward the mesenchymal fate. As evidenced by a recent longitudinal analysis of patient-derived single and clustered CTCs, the number and size of CTC clusters add additional prognostic value to single CTCs’ enumeration alone [114]. Nevertheless, the mechanisms involved in the generation of a certain number and/or size of CTC clusters are yet to be studied. Interestingly, recent findings in this direction point to CTC plasticity as a key regulator of CTC-cluster size. Indeed, the prevention of a full EMT transition and thus a hybrid epithelial/mesenchymal phenotype, regulates the formation of large CTC clusters, suggesting that the balance between intermediate epithelial/mesenchymal phenotypes improved the metastatic fitness of CTC clusters [115].

#### 3.2.2. Interactions That Matter: Heterotypic Clustering

The metastatic fitness of CTCs may be regulated by their physical and functional interactions with cell types other than cancer cells established at the primary tumor site or during their transit through the circulation, thus creating not only polyclonal but also heterotypic clusters (Figure 2). These heterotypic CTC clusters can include neutrophils [79,116], dendritic cells [117], or cancer-associated fibroblasts derived from the primary tumor stroma [118] that accompany CTCs to their secondary site [119]. Those companions are likely to modify the phenotype and intravascular behavior of CTCs by diverse means, including enhanced resistance to shear stress, EMT/MET induction, adhesion, survival, or proliferation. Moreover, the variety of cytokines and growth factors arising from those heterotypic CTC clusters may play a fundamental role in remodeling the distant niche during and after extravasation, thereby facilitating colonization [119,120]. One well-studied heterotypic interaction is that of CTCs and blood platelets (Figure 3). The implication of blood platelets in cancer is a rather old song [121,122]; however, their role in metastasis is not yet completely understood.

In general terms, platelets have shown a pro-metastatic role in several mouse models [86,123,124,125], and their number, size, and thrombotic properties have been linked to poor prognosis in human cancers [83,84,126]. The most compelling evidence for pro-metastatic platelets is the inhibition of metastasis by platelet depletion in experimental murine lung metastasis models [122,127,128]. Additionally, it is generally accepted that CTCs are able to bind, activate, and aggregate platelets in a process called tumor cell-induced platelet aggregation (TCIPA) [129]. TCIPA has been shown to correlate with the metastatic potential of CTCs [130,131] and to protect CTCs from shear stress and/or immune system cytotoxicity by forming a physical shield or by releasing immunosuppressive molecules [86,132]. The mechanism(s) involved in TCIPA-metastatic potential correlation are not yet clear, as not all metastatic cells aggregate platelets [129,133,134,135]. In that sense, TCIPA involvement in metastatic potential may have historically suffered from a lack of consensus about what TCIPA actually is: The induction of homotypic clumps of activated platelets, or the formation of platelet–tumor cell heterotypic clusters? In the later scenario, resting or low-activated platelets could bind and shield cells without classic TCIPA occurrence (unpublished observation). Additionally, and contrary to their well-established pro-metastatic role, specific platelets receptors have been shown to mediate anti-metastatic effects [136,137], questioning their precise contribution to metastasis and suggesting a spatiotemporal role of platelets in the metastatic cascade [138]. Nonetheless, heterotypic interactions are likely to prove a key component of metastatic success and may be refined in the future to include platelet binding to specific CTC populations where adhesive capacity is enhanced, and platelet-dependent tuning of CTC–endothelial adhesion/extravasation.

Whether the effects of platelets on metastasis involve physical and continuous CTC–platelet interactions whilst in the circulation and/or during extravasation remains an open question. Steric interference of CTC–platelet interactions directed at the alpha2beta3 integrin expressed on platelets does inhibit metastatic burden [139,140,141]. On the CTC side, the adhesion protein CD97 that is expressed in several primary and metastatic cancers [142] has been shown to directly interact and activate platelets. In turn, the lysophosphatidic acid (LPA) released by the platelets promotes experimental metastasis [143,144] by a mechanism involving CD97–LPAR (LPA receptor) dimerization at the CTC plasma membrane. LPA binding to CD97–LPAR heterodimer may also induce a pre-EMT invasive phenotype via a RHO family GTPase signaling-dependent mechanism [145,146]. Other cell surface antigens expressed on tumor cells can serve as adhesive receptors for platelets, including podoplanin: CLEC2 [147], the HMGB1: TLR4 [148], and the CD24: P-selectin interactions [149]. Interestingly, CD24 knockdown decreases metastatic burden in vivo, whether this is due to changes in platelet interactions remains to be determined [150,151]. Platelets may also support CTC survival and subsequent metastasis by inhibiting anoikis in a Yes associated protein 1 (YAP1)-dependent manner [152]. They can also tune endothelial fitness and favor adhesion to the vessel wall by activating the purigenic receptor, P2Y_2_ [153,154] or by their natural ability to link to endothelial selectin P ligand (PSGL-1) [155,156], highlighting the important spatiotemporal role of platelets during the metastatic cascade.

#### 3.2.3. Going with the Flow: Biomechanics of CTCs Extravasation

In order to reach secondary sites, circulating tumor cells (CTCs) have to avoid the hostile blood or lymphatic flow forces to arrest and stably adhere to the endothelium of the target organ [157,158]. CTC-extrinsic mechanisms such hemodynamic forces have been proven to be key in CTC endothelial arrest and extravasation [43,159]. We have recently identified a threshold of hemodynamic forces that allow stable arrest of CTCs in low-flow venous-like vascular regions, and active endothelial remodeling in higher-flow regions. Endothelial remodeling is an essential event for successful CTC extravasation (Figure 3). In this sense, endothelial fitness and crosstalk with CTCs at the extravasation site may define the final metastatic outcome. Indeed, we have observed that only flow-activated endothelium shows plasma membrane protrusions and accomplishes endothelial remodeling in vitro [159]. Others have additionally demonstrated that flow forces are able to regulate endothelial cell barrier function via non-canonical Notch signaling [160], making endothelial cells more permeable to CTCs, and inducing vascular cell adhesion molecule 1 (VCAM1) expression, leading to increased neutrophil infiltration and metastasis [161]. Interestingly, the areas with endothelial remodeling show deposition of fibrillar material and platelet recruitment [162,163], supporting a role for platelets in CTC flow-dependent adhesion and/or extravasation processes. Whether CTC clusters equally extravasate by endothelial remodeling in flow-permissive regions remains to be further elucidated. A recent study conducted in the zebrafish embryo, that requires further validation, demonstrates that clusters of CTCs mostly extravasate upon endothelium remodeling [164]. It has become evident that clustering increases CTC resistance to shear stress and protects from immune cell clearance [103,165]. Furthermore, the trajectories traveled by CTC clusters in the circulation are different to the paths of single CTCs, in large part due to size and shape. Compact clusters flow closer to the endothelial barrier than linear clusters or single CTCs and, thus, slowly [166,167], which increases their ability of adhering to the endothelium [168]. Interestingly, the intrinsic differentiation state of a CTC cluster may also influence its flow-dependent adhesive and biomechanical properties. It has been demonstrated that breast cancer cells showing the stem-like CD44^+^/CD24^−^/ALDH1^+^ phenotype were significantly more deformable than non-CSCs. In addition, more-deformable cells were found to roll with shear-independent velocities in vitro [169]. Those findings have provided motivation to consider mechanical properties as a possible biomarker for cancer cell stemness. Indeed, we have recently shown that CD44 plays a key role in early endothelial arrest, as CD44 mediates the early weak-magnitude adhesion forces required for CTC arrest at the endothelial wall [43]. Hence, the increased metastatic potential of CTC clusters could be explained in part by a higher propensity to arrest on endothelial cells and to extravasate, which might be directly linked to the cell deformability index of CTC clusters.

## 4. Secondary Organ Colonization: Shedders or Seeders?

Not all CTCs that reach a secondary site have the capacity to colonize it [170]. In an elegant study utilizing barcoding clonal analysis of patient-derived xenografts, Merino et al. recently demonstrated that the extent of clonal diversity at metastatic sites is highly dependent on continual shedding of CTCs from the primary tumor. Hence, once the primary tumor is removed, clonal heterogeneity in secondary organs is dramatically reduced [18]. It is thus possible that while a variety of heterogeneous cancer cells may continually enter the circulation via active or passive processes at the primary tumor site, only CTCs with tumor-initiating potential have the ability to efficiently seed metastases. This idea is in line with the observation that cancer cells with high tumor-initiating/metastatic potential, for example CD44^Hi^ cells, are observed at low frequency in patient primary tumors (~15% of the cancer cell population [6]). 

There are likely other mechanisms by which poorly metastatic cells overcome their own metastatic inefficiency, including that of subclonal co-operation. It has recently been shown that minor subclones expressing interleukin 11 (IL11) and vascular endothelial growth factor D (VEGFD) within the primary tumor can modulate the immune system in a manner that enhances polyclonal metastatic growth of otherwise non-metastatic clones [171]. Those findings demonstrate how intra-tumor heterogeneity can mechanistically progress disease to advanced stages [172] and highlight the complex and co-operative interactions that contribute to metastatic success. 

### Metastatic Niche: A Driving Force or a Barrier?

Irrespective of whether cancer cells arrive at the secondary tumor site as single, clusters, or polyclonal clusters of cells, there are still multiple extrinsic stresses that must be overcome in order to generate a robustly growing metastasis (Figure 4). In 1889, Steven Paget proposed that the ability of tumor cells to initiate secondary tumor growth largely depends on crosstalk between metastatic tumor cells—the seeds—and the host microenvironment—the soil [173]. For a cancer cell entering a secondary tissue, it is likely that the growth-supportive signals from the local stroma and interactions with other cancer cells are quite different to those formerly present at the primary tumor site. Consequently, even a metastasis-competent disseminated cancer cell may be forced into a state of senescence, apoptosis, or latency if it is not able to rapidly adapt to its new environment. The fate of a disseminated cancer cell at the secondary site can be markedly influenced by location, where proximity to the microvasculature niche is related to dormancy, an effect mediated by tissue specific mechanisms. For example, cancer cell quiescence in the lung is mediated by thrombospondin 1 (TSP1) and bone morphogenetic protein 4 (BMP4), whereas in the bone marrow, TSP1, BMP7, transforming growth factor β2 (TGFβ2), and growth arrest-specific 6 (GAS6) induce and maintain quiescence [174,175].

Metastatic colonization can also be restricted by the immune system. In melanoma, disseminated tumor cells enter into an immuno-induced dormant state following arrival at a secondary site [176]. Upon depletion of cytotoxic CD8^+^ T cells however, metastatic growth reactivates, indicating an active role of the immune system in inhibiting tumor cell proliferation after dissemination. Similarly, in breast and lung carcinomas, interaction between tumor cells and natural killer (NK) cells suppress NK cell-activating ligands, a process that appears to be coupled with entrance into a quiescent state [177]. Our recent work demonstrated that the immune system can restrict metastatic growth by modulating a cancer cell’s phenotype. In models of breast cancer, the primary tumor activated the innate immune system such that macrophages at sites of metastasis inhibit metastatic outgrowth by locking cancer cells in a stem-like state [178]. In that setting, interleukin 1 beta IL1B released by macrophages signals via the interleukin 1 receptor type 1 (IL1R1) on the cancer cells to maintain high expression of ZEB1. Implicit here is the notion that preventing CD44^Hi^ cells from undergoing asymmetric divisions to produce their highly proliferative epithelial CD44^Lo^ progeny significantly inhibits metastatic growth. Conversely, forcing cells to undergo a complete mesenchymal-to-epithelial transition (MET) at the metastatic site can deplete the tumor of the tumor-initiating cells that sustain secondary tumor growth [179]. Together those studies highlight the intricate balance between epithelial and mesenchymal cancer cell states and their impact on tumorigenicity.

In other instances, including the example of co-operative growth leading to robust metastasis [171], the immune system can act in a manner that enhances secondary tumor growth. In that setting, a hypoxic primary tumor microenvironment creates a pre-metastatic niche comprising CD11b^+^/Ly6C^med^/Ly6G^+^ immune suppressor cells that compromise NK cell cytotoxicity, thereby diminishing a key mechanism for disseminated tumor cell elimination [180]. Additionally, recruitment of monocytes/macrophages and neutrophils can promote tumor cell survival, colonization, and pre-metastatic niche establishment in mice [181,182,183]. Neutrophils have been shown to enhance metastasis by grouping CTCs in circulation through the formation of neutrophil traps (NETs)—nets of extracellular neutrophil DNA fibers. In addition, neutrophil-derived leukotrienes were shown to be responsible for colonization at sites of metastasis by selectively expanding a subpopulation of cancer cells that retain high tumorigenic potential. Neutrophils can also remodel the host extracellular matrix to promote metastatic growth and direct signaling that maintains aggressive metastasis-initiating phenotypes [171,182,183,184].

The EMT itself is another mechanism that can impart several advantages on disseminated tumor cells during early stages of colonization. Tumor cells are subjected to high levels of oxidative stress due to hypoxic conditions at the primary tumor site, in the circulation, and at secondary sites of colonization [185]. Expression of EMT transcription factors can protect from the damaging cytotoxic effects induced by oxygen radicals and DNA damage [186]. Furthermore, oxidative stress has been linked with activation of the EMT [187], setting up a positive feedback loop that may enhance metastatic cell survival under stressful conditions. The EMT also plays a key role in enabling disseminated tumor cells to evade immune surveillance [188]. Accordingly, epithelial cells have been shown to express high major histocompatibility (MHC) class I and low CD274 (PD-L1) levels, while more mesenchymal carcinoma cell lines exhibiting EMT markers expressed low levels of MHC-I, high levels of PD-L1 [189]. Indeed, ZEB1 can directly regulate PD-L1 levels [190,191]. Consequently, epithelial tumors can be more susceptible to elimination by immunotherapy than corresponding mesenchymal tumors [189]. Mechanisms of immune evasion attributable to the EMT may also include downregulation of immunoproteasome subunits and consequently, downregulation of MHC class I -bound peptides [192].

Another important determinant of metastatic success is the preparation of a favorable pre-metastatic niche via primary tumor cell-derived extracellular vesicles (EV) [193]. A recent study by Hoshino et al. showed that uptake of tumor-derived integrin exosomes by resident cells at secondary sites determines organotropic metastasis. Exosomal expression of α6β4 and α6β1 is associated with lung metastasis, while exosomal integrin αvβ5 was linked to liver metastasis. Reduction of those distinct integrin complexes decreased exosome uptake and subsequently metastasis, via inhibition of Src signaling and activation of pro-inflammatory signals in resident cells [194]. Additionally, our recent work has shown that patrolling monocytes and endothelial cells are key cellular types in charge of tumor EV uptake [195] and may therefore be early activators of the metastatic niche. Indeed, uptake of metastatic tumor cell-derived molecules reprograms the resident normal tissue cells in a manner that aids metastatic growth [196]. Clearly, interactions between the seed and soil are intricately linked to metastatic success. Determining the mechanisms that define those interactions may form the basis of future therapeutic strategies to inhibit metastasis.

## 5. Conclusions

Metastasis is not a linear process, rather, it is a highly dynamic interplay of intrinsic cellular properties and extrinsic host factors that are constantly evolving throughout the course of tumorigenesis to positively or negatively influence the metastatic process. We have discussed phenotypic traits that promote a cancer cell’s ability to complete specific stages of the metastatic cascade, encompassing the notion that there is not a single phenotypic state that equates with metastatic success. Instead, it is likely that metastatic success lies in tumor cell’s ability to adapt its phenotype, at each step of the cascade, to survive the variety of challenges encountered along its journey; including constant turnover of transitional cellular states, interactions with host components and between different clonal populations. The extent to which a given cell/clone completes the metastatic cascade likely depends upon its epigenetic, transcriptomic, and proteomic landscape, and whether it travels alone or with companions. Those properties, in turn, determine how that cell processes and responds to incoming signals. In some tumor types, it is likely that specialized cancer cells are equipped with most, if not all of the biological traits required for metastasis. In less adept cancer cell populations, a favorable metastatic niche environment, traveling with a support team, or a permissive environment created by the primary tumor may be the prime determinants of metastatic success. Elucidating the prominent mechanisms at play in different tumor types and subtypes will lead to more effective means to therapeutically target and inhibit metastatic growth.

## Figures and Tables

**Figure 1 cancers-11-01575-f001:**
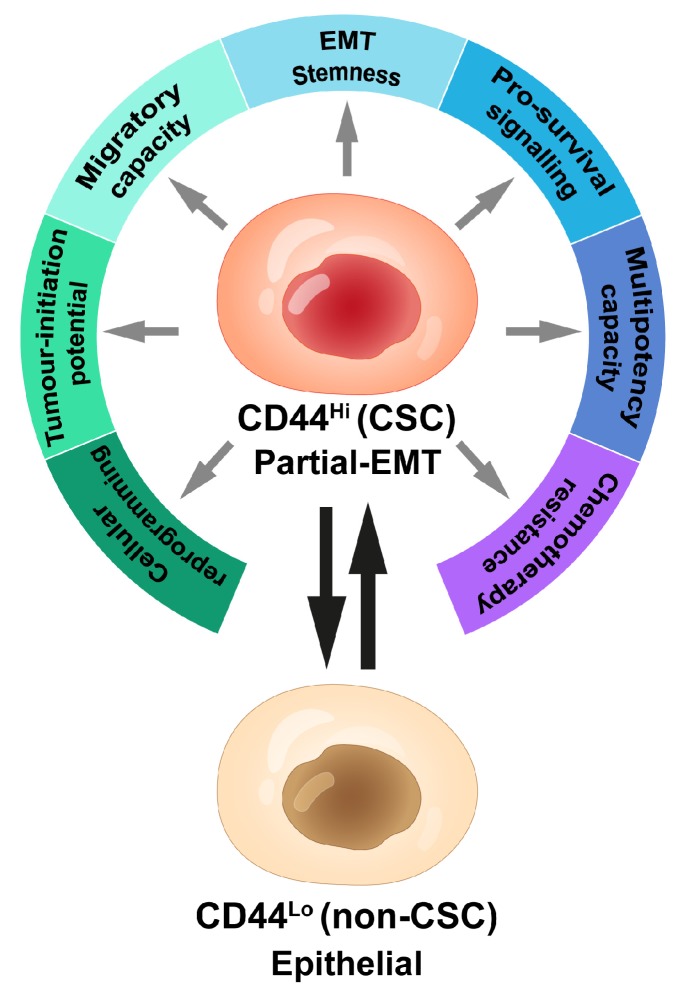
Schematic overview of cancer cell phenotypic heterogeneity. Cancer cells with an identical genetic background can be hierarchically organized according to cell phenotype. CD44^Hi^ cancer stem cells (CSC) are an aggressive cell type that have likely undergone a partial-epithelial-to-mesenchymal transition (partial-EMT) to acquire multiple biological traits that enhance their tumorigenic and metastatic potential. Cells residing in a CD44^Hi^ CSC state sit at the top of the hierarchy, where they can self-renew to maintain the aggressive CSC pool or, alternatively, undergo asymmetric divisions to form more differentiated CD44^Lo^ (non-CSC) progeny. In some cancer types, CD44^Lo^ epithelial cells have the potential to ascend the hierarchy and enter into the aggressive CD44^Hi^ state.

**Figure 2 cancers-11-01575-f002:**
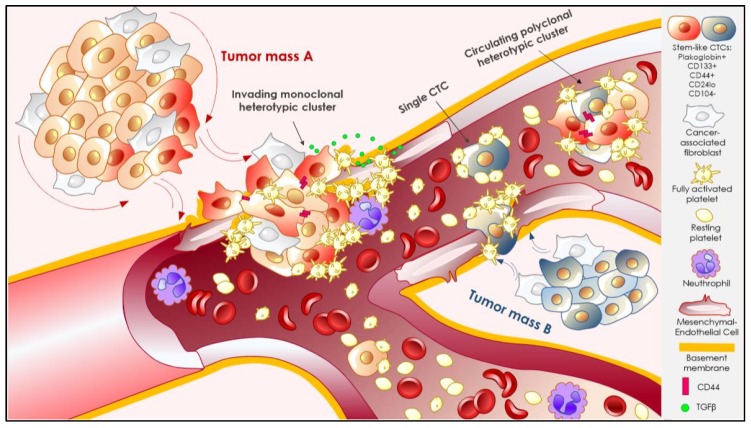
From invasion into the circulation. Tumor cells can reach the vasculature and enter the circulation as single circulating tumor cells (CTCs) or CTC clusters. The latter may show a variable degree of complexity according to cell heterogeneity within the primary tumor (tumor mass A) and/or the cells encountered during the process of intravasation and in the circulation, such as blood cells (e.g., platelets, neutrophils) or due to encounters with tumor cells from a different primary site (tumor mass B). Cancer cells within the primary tumor can reside in diverse stages of differentiation along an epithelial-to-mesenchymal spectrum. Cells that display mesenchymal features may have enhanced survival, proliferation, and invasiveness and express cancer stem-like markers, including the adhesion molecules CD44 or plakoglobin. Homotypic interactions between tumor cells, mediated by CD44 among others, may lead to the formation of a CTC cluster. At the moment of intravasation, disruption of endothelial integrity by invasive tumor cells exposes extracellular matrix proteins (yellow line) including von Willebrand factor (vWF), collagen, or fibronectin, which recruit and activate blood platelets. In turn, platelets secrete transforming growth factor beta TGFβ, among many other angiogenic and pro-inflammatory factors that can induce tumor cells to undergo the EMT and induce a mesenchymal phenotype in endothelial cells, thereby increasing endothelial permeability and the expression of Notch ligands. Activation of Notch signaling in tumor cells supports survival and proliferation, mostly on CSC populations. Once tumor cells have entered the circulation, activated or resting platelets (unpublished observation) can bind to single CTCs or CTC clusters and support survival by protecting them from shear stress as well as enhancing cell adhesion at distant sites of arrest.

**Figure 3 cancers-11-01575-f003:**
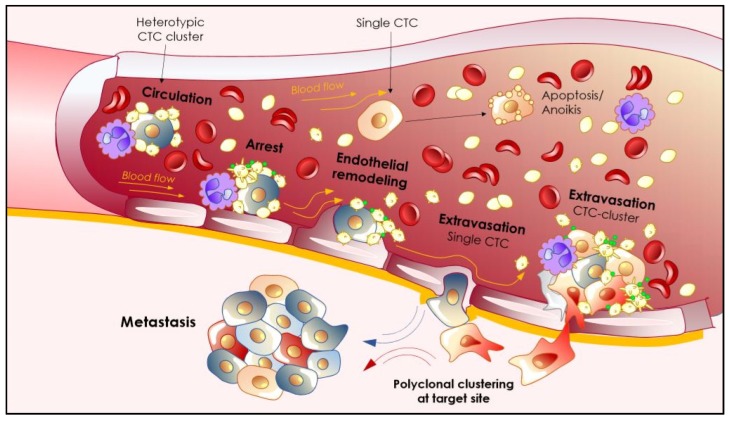
From the circulation to extravasation. CTCs that survive the harsh conditions of the blood microenvironment will eventually come into contact with, and arrest, on the endothelial cells lining the blood vessels at the metastatic site. Adhesion to the endothelial cells depends on the adhesion receptor repertoire of the tumor cells and in the case of heterotypic clusters, on the adhesion receptor repertoire of accompanying cells, for example, neutrophils and platelets. In addition to their role in adhesion, neutrophils and platelets can further enhance extravasation by increasing endothelial permeability via TGFβ and vascular endothelial growth factor A (VEGF secretion. Endothelial arrest predominantly takes place at sites where blow flow is low enough to allow stable adhesion to the vasculature. After this point, higher flow profiles are needed to induce endothelial remodeling around the arrested CTC, an essential process for successful extravasation. Clustering of polyclonal CTCs can occur at the site of arrest and/or extravasation, together with blood cells.

**Figure 4 cancers-11-01575-f004:**
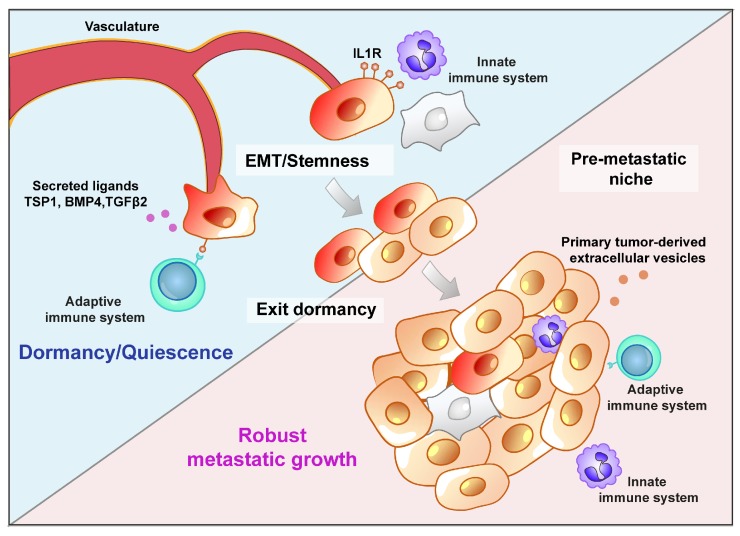
Metastatic colonization. Cancer cells with the intrinsic potential (e.g., CSCs) to initiate a secondary tumor must overcome multiple extrinsic stresses to establish a robustly growing metastasis. Signaling in the secondary tumor environment (initiated by the innate/adaptive immune system, stromal cells, or vasculature) can induce cells into a state of quiescence or dormancy. A permissive pre-metastatic niche may be created by signals arising from the primary tumor (via primary tumor-derived extracellular vesicles or polarization of the adaptive and/or innate immune systems) that enables cancer cells to avoid or exit dormancy and subsequently proliferate to establish a metastatic colony.

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
