# Peer review of "The Complexities of Metastasis"

_cancers, 2019, doi:10.3390/cancers11101575_

Round 1

Reviewer 1 Report

The authors have reviewed the literature about recent updates in decoding the mechanisms underlying various steps of the metastatic-invasion cascade. While the manuscript covers a timely theme to review, it misses discussing about many latest and crucial observations, particularly at a mechanistic level, including the questions which are being most actively debated currently. Thus, the authors should address the below mentioned comments before the manuscript can be recommended for publication:

Section 1.3.1 – The authors mention that “EMT signature also correlates with poor clinical outcome”. This view is outdated, as recent preclinical and clinical evidence suggests that the correlation between EMT and aggressiveness/clinical outcome is not a monotonic one, i.e. more EMT does not always lead to poorer outcomes; in fact, it may be associated with a better outcome at times. Instead, one or more hybrid E/M phenotypes seem to be the most aggressive. This association between EMT and survival has not been universally seen in two EMT scoring methods developed recently (Tan et al. EMBO Mol Med 2014; George et al. Cancer Res 2017). Recent clinical data also suggests that hybrid E/M CTCs may be a better indicator of patient survival (Sun et al. EBioMedicine 2019).

Section 1.3.2 – The authors mention that “partial EMT state provides optimal tumorigenicity”. Some most relevant references and pioneering studies that presented evidence for the same before it became the trend are not cited – Grosse-Wilde et al. PLoS ONE 2015; Jolly et al. J R Soc Interface 2014; Pastushenko et al. Nature 2018.

Section 1.3 – The authors mention about CD44 isoform variants being associated with several features; however, the mechanistic underpinning of why/how CD44 isoform variants show that trend is not discussed. ZEB1 – a key EMT inducer discussed later in the review – can drive the alternative splicing of CD44 isoforms (Preca et al. Oncotarget 2017; Jolly et al. APL Bioeng 2018). The authors should briefly comment on the same.

Section 3 – The authors mention that “studies analyzing CTCs in human patients are enriched for an EMT phenotype”. First, the referenced paper does not necessarily claim the same, instead it mentions about the heterogeneity in the EMT status of CTCs. Second, the necessity of (at least a complete) EMT for metastasis has been challenged recently (Zheng et al. Nature 2015; Fischer et al. Nature 2015; Padmanabhan et al. Nature 2019) and active debate is ongoing (Krebs et al. Nat Cell Biol 2017) to identify confounding factors that are puzzling the case. A reconciliatory attempt has been offered from a systems biology perspective (Jolly et al. Mol Oncol 2017); the authors should spend some time discussing this important aspect which remains an open question.

Section 3.2.1 – The authors mention that “the mechanisms behind a CTC clusters’ enhanced metastatic fitness are currently under investigation”. While many important points are discussed, one additional point could be the possibility of the passive migration of clusters as suggested by some pathologists – Bronsert et al. J Pathol 2014; Fang et al. Hepatology 2015.

Section 3.2.1 – The authors mention that “the number and size of CTC clusters add additional prognostic value to single CTCs enumeration alone”. A more systematic attempt to understand the frequency and size distributions of CTC clusters was recently made (Bocci et al. Cancer Res 2018) which is worth discussing.

Section 4.2 – The authors mention about “EMT also plays a key role in enabling disseminated tumor cells to evade immune-surveillance”. They should consider discussing these studies which offer mechanistic insights into the phenomenological observations discussed here – Tripathi et al. PNAS 2016; Chen et al. Nat Comm 2014; Noman et al. Oncoimmunology 2017.

Section 4.3 – The authors mention that “it is likely that metastatic success lies in tumor cell’s ability to adapt their phenotype to survive the variety of challenges encountered along its journey”. This is an important point, suggesting the contributions of phenotypic plasticity instead of mutational signatures for metastasis. A recent review highlighted this point that no unique mutational signatures have yet been identified for metastasis (Celia-Terrassa & Kang, Genes Dev 2016). How do the authors state their position vis-à-vis this observation?

Section 1.2 – The authors discuss about the role of epigenetic mechanisms in generating phenotypic heterogeneity. While an important mechanism, epigenetic changes is not the only way in which phenotypic heterogeneity can be generated. The authors should briefly discuss about other mechanisms underlying phenotypic (or non-genetic) heterogeneity such as a) biological noise that can generate spontaneous transitions among phenotypes (Ruscetti et al. Oncogene 2016; Tripathi et al. bioRxiv 2019; Bhatia et al. J Clin Med 2019), b) asymmetry in EMT/MET that can generate different subsets of cells in high-dimensional EMT landscape (Karacosta et al. bioRxiv 2019), thus generating phenotypic heterogeneity.

Author Response

We thank Reviewer 1 for their helpful feedback. Please find below detailed responses to the queries raised.

The authors have reviewed the literature about recent updates in decoding the mechanisms underlying various steps of the metastatic-invasion cascade. While the manuscript covers a timely theme to review, it misses discussing about many latest and crucial observations, particularly at a mechanistic level, including the questions which are being most actively debated currently. Thus, the authors should address the below mentioned comments before the manuscript can be recommended for publication:

Section 1.3.1 – The authors mention that “EMT signature also correlates with poor clinical outcome”. This view is outdated, as recent preclinical and clinical evidence suggests that the correlation between EMT and aggressiveness/clinical outcome is not a monotonic one, i.e. more EMT does not always lead to poorer outcomes; in fact, it may be associated with a better outcome at times. Instead, one or more hybrid E/M phenotypes seem to be the most aggressive. This association between EMT and survival has not been universally seen in two EMT scoring methods developed recently (Tan et al. EMBO Mol Med 2014; George et al. Cancer Res 2017). Recent clinical data also suggests that hybrid E/M CTCs may be a better indicator of patient survival (Sun et al. EBioMedicine 2019).

We agree with Reviewer 1 that the hybrid- or partial-EMT state may be a better indicator of aggressiveness than a complete mesenchymal state. Indeed, that is the exact sentiment that we have captured and illustrated in Figure 1, and the topic of discussion in the following section -1.3.2.  With that said, we do agree with Reviewer 1 and do not wish to imply that an EMT signature always correlates with poor clinical outcome. Accordingly we have qualified our statement at line 109 to read: Moreover, loss of an epithelial phenotype and gain of mesenchymal features correlates with poor clinical outcome in some tumor types[52-57]. We have also included the Grosse-Wilde PLoS One reference.

Section 1.3.2 – The authors mention that “partial EMT state provides optimal tumorigenicity”. Some most relevant references and pioneering studies that presented evidence for the same before it became the trend are not cited – Grosse-Wilde et al. PLoS ONE 2015; Jolly et al. J R Soc Interface 2014; Pastushenko et al. Nature 2018.

We have included the references cited above, and in addition, Jolly MK, Pharmacol Ther, 2019, at line 126.

Section 1.3 – The authors mention about CD44 isoform variants being associated with several features; however, the mechanistic underpinning of why/how CD44 isoform variants show that trend is not discussed. ZEB1 – a key EMT inducer discussed later in the review – can drive the alternative splicing of CD44 isoforms (Preca et al. Oncotarget 2017; Jolly et al. APL Bioeng 2018). The authors should briefly comment on the same.

We have included a comment on this at line 98: The EMT transcription factors Snail1, Snail2, Zeb1, among others, are key mediators of that process[46]. Indeed, ZEB1 also drives splicing of CD44 in a manner that promotes tumorigenicity, recurrence and drug-resistance [47-48].

Section 3 – The authors mention that “studies analyzing CTCs in human patients are enriched for an EMT phenotype”. First, the referenced paper does not necessarily claim the same, instead it mentions about the heterogeneity in the EMT status of CTCs. Second, the necessity of (at least a complete) EMT for metastasis has been challenged recently (Zheng et al. Nature 2015; Fischer et al. Nature 2015; Padmanabhan et al. Nature 2019) and active debate is ongoing (Krebs et al. Nat Cell Biol 2017) to identify confounding factors that are puzzling the case. A reconciliatory attempt has been offered from a systems biology perspective (Jolly et al. Mol Oncol 2017); the authors should spend some time discussing this important aspect which remains an open question.

We have further clarified our opinion on the phenotypic state of CTCs at line 257 in the following manner: However, the specific role of EMT in CTC cluster formation and the resultant enhanced metastatic fitness remains unclear. For example, it has been shown that CTC-clusters encapsulated by tumor-induced blood vessels are highly metastatic by a Slug/Snail-independent mechanism[108]. Furthermore, another report by using quantitative 3D histology at the cancer-host interface revealed that collective migration is the predominant mechanism of cancer cell invasion, positioning single cell migration as an extremely rare event [109]. These findings suggest that CTC-extrinsic mechanisms, such as vascular patterning during tumor progression, can influence CTC clustering and shedding without a compulsory phenotypic change towards the mesenchymal fate.

With respect to the debate surrounding the relevance of EMT, in our opinion, the two papers mentioned by the reviewer have extensive scientific and experimental flaws. Accordingly, we do not believe that the conclusions made by Zheng et al, and Fisher et al, are correct. Our opinion on this matter is echoed by the commentary: Upholding a role for EMT in pancreatic cancer metastasis, Nicole M. Aiello, Thomas Brabletz, Yibin Kang, M. Angela Nieto, Robert A. Weinberg & Ben Z. Stanger, Nature, 2017.  As such, we have chosen to not dwell on that issue as it is poorly substantiated. We do however believe that the EMT is not the only mechanism by which cancer cells acquire a metastatic phenotype. As stated in the abstract at line 20: Phenotypic heterogeneity can be derived from a combination of factors, in which the genetic make-up, interaction with the environment and ability of cells to adapt to evolving microenvironments and mechanical forces play a major role. Furthermore, we highlight in paragraph 1.1, line 31, that there are multiple factors that contribute to tumor progression, including genetic and epigenetic mechanisms.

Section 3.2.1 – The authors mention that “the mechanisms behind a CTC clusters’ enhanced metastatic fitness are currently under investigation”. While many important points are discussed, one additional point could be the possibility of the passive migration of clusters as suggested by some pathologists – Bronsert et al. J Pathol 2014; Fang et al. Hepatology 2015.

We have added discussion on this point at line 260.

Section 3.2.1 – The authors mention that “the number and size of CTC clusters add additional prognostic value to single CTCs enumeration alone”. A more systematic attempt to understand the frequency and size distributions of CTC clusters was recently made (Bocci et al. Cancer Res 2018) which is worth discussing.

We have added discussion on this point at line 269. 

Section 4.2 – The authors mention about “EMT also plays a key role in enabling disseminated tumor cells to evade immune-surveillance”. They should consider discussing these studies which offer mechanistic insights into the phenomenological observations discussed here – Tripathi et al. PNAS 2016; Chen et al. Nat Comm 2014; Noman et al. Oncoimmunology 2017.

We have included discussion on these points and references at lines 439 - 443

Section 4.3 – The authors mention that “it is likely that metastatic success lies in tumor cell’s ability to adapt their phenotype to survive the variety of challenges encountered along its journey”. This is an important point, suggesting the contributions of phenotypic plasticity instead of mutational signatures for metastasis. A recent review highlighted this point that no unique mutational signatures have yet been identified for metastasis (Celia-Terrassa & Kang, Genes Dev 2016). How do the authors state their position vis-à-vis this observation?

We have discussed the idea about clonal drivers of primary and metastatic growth in section 4 where we consider ‘shedders’ versus ‘seeders’. The focus of this review is really entirely on that notion that inspite of genetic variation, there will always be some level of epigenetic regulation, that is to say, phenotypic plasticity, contributing to metastatic success. As such, our opinions are in line with the CeliaTerrassa and Kang statement on the lack of unique mutational signatures yet to define metastasis. We have now pointed to that review at line 360.

Section 1.2 – The authors discuss about the role of epigenetic mechanisms in generating phenotypic heterogeneity. While an important mechanism, epigenetic changes is not the only way in which phenotypic heterogeneity can be generated. The authors should briefly discuss about other mechanisms underlying phenotypic (or non-genetic) heterogeneity such as a) biological noise that can generate spontaneous transitions among phenotypes (Ruscetti et al. Oncogene 2016; Tripathi et al. bioRxiv 2019; Bhatia et al. J Clin Med 2019), b) asymmetry in EMT/MET that can generate different subsets of cells in high-dimensional EMT landscape (Karacosta et al. bioRxiv 2019), thus generating phenotypic heterogeneity.

As stated in the abstract, we acknowledge that there are clearly other mechanisms that contribute to heterogeneity, of which not all can be adequately covered in this review. We have indeed covered symmetric and asymmetric division with respect CD44Lo and CD44Hi interconversions (section 1.2), and the role of EMT and specifically MET at the metastatic site (section 4.1 line 413).

Reviewer 2 Report

This paper is an extensive and well documented review of the metastatic process, with an overview on primary tumors, circulating tumor cells and metastases. I have only a remark on the behaviour of metastatic tumor cells which may express a mesenchymal to epithelial transition (MET) in the metastatic site, which is not reported in this paper.

There also numerous typing mistakes in the text to be corrected.

Author Response

We thank Reviewer 2 for their feedback.

We have corrected the spelling issues (apologies! we clearly had issues with Word spell-check switching between the french and english versions!)

We have added a comment with respect to the importance of MET at the metastatic site and have referenced this accordingly (line 415). 

Round 2

Reviewer 1 Report

The authors have now addressed the comments raised in a satisfactory way.

Author Response

We acknowledge that we are addressed all of Reviewer 2's concerns.